# Flexible Attachment and Detachment of Centromeres and Telomeres to and from Chromosomes

**DOI:** 10.3390/biom13061016

**Published:** 2023-06-20

**Authors:** Riku Kuse, Kojiro Ishii

**Affiliations:** Laboratory of Chromosome Function and Regulation, Graduate School of Engineering, Kochi University of Technology, Kochi 782-8502, Japan

**Keywords:** telomere fusion, centromere inactivation, telomere healing, neocentromere formation

## Abstract

Accurate transmission of genomic information across multiple cell divisions and generations, without any losses or errors, is fundamental to all living organisms. To achieve this goal, eukaryotes devised chromosomes. Eukaryotic genomes are represented by multiple linear chromosomes in the nucleus, each carrying a centromere in the middle, a telomere at both ends, and multiple origins of replication along the chromosome arms. Although all three of these DNA elements are indispensable for chromosome function, centromeres and telomeres possess the potential to detach from the original chromosome and attach to new chromosomal positions, as evident from the events of telomere fusion, centromere inactivation, telomere healing, and neocentromere formation. These events seem to occur spontaneously in nature but have not yet been elucidated clearly, because they are relatively infrequent and sometimes detrimental. To address this issue, experimental setups have been developed using model organisms such as yeast. In this article, we review some of the key experiments that provide clues as to the extent to which these paradoxical and elusive features of chromosomally indispensable elements may become valuable in the natural context.

## 1. Introduction

Centromeres, telomeres, and origins of replication are the three major DNA elements that enable linear DNA molecules in eukaryotes to function as chromosomes [1]. In the cell nucleus, these DNA elements independently accommodate a wide variety of protein machineries all dedicated to faithful genome inheritance through the accurate duplication and segregation of entire chromosomes [2,3,4,5,6,7]. Since these three DNA elements display no overlapping or redundant functions, none of them can be lost from the chromosomes. However, all three elements curiously exhibit a certain level of ambiguity in their state of existence. For example, centromeric regions in many organisms are epigenetically defined by the accumulation of histone H3 variant CENP-A-containing nucleosomes, locally interspersed with canonical nucleosomes. Epigenetic mechanisms are more plastic and reversible than DNA sequence and reduce the ability of each centromere to remain in a fixed position on the chromosome [8,9]. The repetitive nature of centromeric DNA, known as α-satellite in humans, also decreases the precision of CENP-A nucleosome distribution along the region [10,11]. Telomeric DNA is also repetitive, varies in length, and is intrinsically unstable across generations, not only because telomeres are located at chromosome termini and are subject to an end-replication problem [4,12,13], but also because telomere length is maintained in equilibrium with multiple opposing factors, including telomerases and components of the DNA double-strand break (DSB) repair pathway [5,12,14,15]. In addition, a multitude of replication origins are arranged randomly along the chromosome arms, in many cases with poor DNA sequence conservation [6], and the activation of each of them is neither obligatory for the completion of whole-genome duplication nor synchronous in each S-phase, but is largely influenced by the spatial organization of the nucleus [16,17].

These virtual ambiguities in the chromosomal localization of centromeres, telomeres, and origins of replication suggest that these elements are not deeply rooted in the chromosomal DNA and instead are inherently meant to be lost from chromosomes or to be generated at a new chromosomal position de novo. Given the essential nature of these elements within chromosomes, such an argument for their active detachment and attachment is seemingly irrational. However, this argument may become relevant once the multiplicity of chromosomes is taken into consideration. The nucleus is tolerant of multiple chromosomes of diverse sizes and shapes. Any losses or gains of these elements, especially centromeres or telomeres, may be accommodated by various secondary alterations in chromosome size, shape, and number (Figure 1). These alterations are often catastrophic and are referred to collectively as gross chromosomal rearrangements (GCRs) [18,19]. Detachment and attachment of centromeres and telomeres occurs multiple times during GCRs [10,19,20] (Figure 1). Thus, the flexibility of centromeres and telomeres is at the core, and not merely the initial trigger, of catastrophes.

GCRs have been frequently observed in malignant tumor cells [10,13,19,20]. Apart from catastrophes, GCR-like chromosomal alterations have also been implicated in reproductive isolation [22,23,24,25]. This suggests that the loose attachment of centromeres and telomeres to chromosomes, whether detrimental or beneficial, is a unique property of eukaryotic chromosomes that cannot be ignored. In this review, we discuss the flexible chromosomal attachment and detachment of telomeres and centromeres. Most of the events discussed in this review are barely detectable under normal cell conditions. Therefore, this review lays a special emphasis on the methodology used to capture the events, with a greater focus on the key experiments performed in various yeast species. In addition, this review does not dive into the mechanistic details of each event but introduces only the bare minimum of what is known. For detailed mechanisms, readers are directed to many excellent reviews on each topic [13,18,26,27,28,29,30,31,32,33,34,35]. Instead, this review sheds light on the natural frequencies of events in the wild-type context. This allows a comparison of the efficiencies of different events in native context and permits combining these efficiencies in the future, which may be useful in obtaining an integrated perspective on chromosome flexibility (Figure 1).

## 2. Spontaneous Telomere Fusion: A Manifestation of Functional Telomere Detachment

The first question to be addressed is the extent to which native telomeres are prone to fusion (Figure 2) under normal conditions. Telomeres are tandemly arranged short DNA repeats, with 3′ single-stranded overhangs, located at the ends of chromosomes, that serve to protect the chromosome ends from nucleolytic degradation and DNA damage responses. The capping function of telomeres is fulfilled through the action of specialized proteins recruited to the repeat sequences, as well as through the formation of a higher order structure [5,14]. Telomere repeats are synthesized by telomerase, a specialized reverse transcriptase that elongates the 3′ end of telomeric DNA using a short region of its RNA subunit as a template. Telomerase action buffers against progressive telomere erosion caused by the incomplete replication of terminal DNA in each cell cycle and sustains the foundation for telomere structure [4,5]. Hence, telomeres are subject to uncapping, and telomere fusions occur readily when the telomere capping protein or the DSB checkpoint signaling along with the telomerase enzyme is functionally compromised [36,37,38,39,40,41,42,43,44]. However, our concern is the occurrence of telomere fusions in an unchallenged, natural context.

### 2.1. Estimation of Spontaneous Telomere Fusion Frequency in Mammals

A PCR-based single molecule assay using the primers designed for telomere-adjacent regions encompassing single-nucleotide polymorphisms and directed toward the chromosome terminus captured telomere fusions at a frequency of ~4 × 10^−6^ in telomerase-positive cultured human cells [45] (Table 1). In this assay, the fusion-committed chromosomal ends exhibited either extensive telomere shortening or no telomeric association, suggesting a stochastic telomere loss as a potential trigger for chromosomal rearrangements in the native context [45,46]. Additionally, experiments using the mouse in vivo model revealed that critically shortened telomeres specifically undergo fusion [47]. In human cells, sequence analysis of telomere fusion junctions indicated that the spontaneous fusion event is mediated by the microhomology-mediated end-joining (MMEJ) pathway of DSB repair [46,48,49]. Importantly, however, complex chromosomal rearrangements, more than just a fusion, were detected in many of the fusogenic clones [46,50], and careful interpretation may be required to understand the consequences of such consecutive complex events [13,51,52] (Figure 1).

### 2.2. Spontaneous Telomere Fusion in Fission Yeast (Schizosaccharomyces pombe)

Telomere fusion is often recognized as a part of consecutive and complex chromosomal rearrangements because it results in the formation of a dicentric chromosome, which leads to chromosome breakage in the following cell division and initiates further rounds of broken-end fusion and dicentric chromosome formation, a phenomenon called the breakage–fusion–bridge (BFB) cycle, originally proposed by McClintock (Figure 1) [13,20,21,72]. However, this scenario applies only to inter-chromosome and inter-sister chromatid fusions, which give rise to dicentric chromosomes; instead, fusion of telomeres within a single chromatid gives rise to a monocentric circular chromosome, which lacks any further connection to the BFB cycle [73]. Taking advantage of the simplicity of intra-chromatid fusions, a smart attempt was made to analyze the naturally occurring telomere fusions in *S. pombe* through circular chromosome formation [56]. In this analysis, a plasmid-originated linear minichromosome [74] with a split marker gene integrated at both telomeric ends was prepared and used to detect the occurrence of its spontaneous circularization by positive genetic selection [56] (Figure 3).

The frequency of minichromosome circularization in wild-type *S. pombe* was estimated as ~1 × 10^−4^ [56] (Table 1), which is 40-fold higher than the frequency of spontaneous telomere fusion in cultured human cells [45]. The characteristics of DNA ends involved in telomere fusions in *S. pombe* were also different from those in normal human cells; the vast majority of the fused ends in *S. pombe* retain many telomere repeats [56]. Minichromosome circularization in wild-type *S. pombe* resembled telomere fusions in human cells expressing a dominant-negative mutant of TRF2, a representative telomere capping protein [36]. The efficiency of chromosome circularization in *S. pombe* was reduced by approximately 10-fold in *lig4* and *pku70* mutants, indicating that chromosome circularization is dependent on the non-homologous end-joining (NHEJ) pathway of DSB repair [48,49,56], which again is in contrast to the MMEJ dependency of the reaction in normal human cells [46] and reminiscent of the fusion observed in TRF2-deficient human cells [75].

These inconsistencies in the characteristics of telomere fusions among different organisms may simply reflect the species-specific regulation of telomere maintenance [4,5,14]. Alternatively, the minichromosomes used in this assay lack centromeres and are therefore maintained unstably as multiple copies (5–7 on average) during cell growth, which may cause the insufficiency of Taz1, a functional TRF2 homolog in *S. pombe*, to a minor fraction of minichromosome telomeres, leading to a TRF2 mutant-like phenotype despite the artificial doubling of the *taz1*^+^ expression level [43,56]. In addition, the telomeres of minichromosomes generated in vivo using a telomere healing-like mechanism (see Section 4) lack the vast majority of subtelomeric sequences that usually exist in regions adjacent to telomeric repeats in the authentic chromosomes of *S. pombe* [74,76], which may confer minichromosome-specific telomere fusion characteristics.

With respect to authenticity, native chromosomes are undoubtedly the best substrate for telomere fusion analysis. However, telomere fusion between native chromosomes is doomed to result in the BFB cycle through dicentric chromosome formation. To overcome this issue, another attempt was made in *S. pombe* to artificially remove the centromere from a native chromosome [77]. In this assay, site-specific recombination was specifically designed between regions located upstream and downstream of a given centromere to concomitantly elicit centromere excision and selectable marker gene expression, allowing the detection of telomere fusion, represented by surviving cells, under specific selection conditions (Figure 3).

By applying the centromere deletion assay to individual chromosomes of *S. pombe*, spontaneous fusions of authentic telomeres were observed at an average frequency of ~6 × 10^−5^ [58] (Table 1). These spontaneous fusions were approximately 2-fold less efficient than minichromosome circularization [56] and occurred equally or even more efficiently in *lig4* and *pku70* mutant backgrounds, suggesting that the mechanism of these fusions was different from the NHEJ-based mechanism of minichromosome circularization [78]. The fusion pattern was also different from that seen in minichromosome circularization but identical to that observed in the telomerase mutant *trt1* and the telomere capping protein mutant *pot1*, in which fusion was observed between hotspots in subtelomeric regions [41,42,79]. However, telomere fusions in the centromere deletion assay were independent of single-strand annealing (SSA), the DSB repair mechanism previously shown to mediate the fusion reaction in *trt1* and *pot1* mutants [48,49,78,79]; instead, they were suggested to be mediated by an illegitimate homologous recombination (HR)-directed rearrangement, which could also induce copy number alteration in the genome upon centromere deletion [78]. Given the relatively frequent de novo formation of ectopic centromeres, also known as neocentromeres (see Section 5), in the subtelomeric regions of centromere-deleted acentric chromosomes in the same assay, the subtelomeric region-mediated fusions detected in this assay may not be truly relevant to the spontaneous telomere fusion observed under normal conditions.

### 2.3. Spontaneous Telomere Fusion in Budding Yeast (Saccharomyces cerevisiae)

In light of the above results, a careful examination of telomere fusion was recently performed in *S. cerevisiae* [61]. The same scheme as the centromere deletion assay in *S. pombe* was applied to wild-type *S. cerevisiae*, and chromosome VI was carefully chosen for the assay, because the immediate lethality of *S. cerevisiae* upon chromosome VI mis-segregation [80] ensured telomere fusion [81]. The zero potential of neocentromere formation in *S. cerevisiae*, owing to the absolute DNA sequence dependency of *S. cerevisiae* centromeres (see Section 5), was also advantageous for telomere fusion-specific selection. In addition, centromere deletion was induced in quiescent cells to facilitate spontaneous telomere fusion [81] and also to avoid the presumed emergence of unusual fusion-prone chromosomes, even at a low frequency, during cell proliferation [56,78].

In this assay, the spontaneous fusion of chromosome VI with other chromosomes was detected at a frequency of ~1 × 10^−7^ [61] (Table 1). The success of detecting such a low frequency event indicated the excellence of this assay, because the previous attempt to detect the same fusion using a transcription-mediated centromere inactivation [82] (Figure 3, see also Section 3) failed in the wild-type background and succeeded only in the telomere capping-protein mutant background [81] or upon exogenous addition of a linker DNA fragment that facilitated telomere fusion in the wild type [83].

The obtained fusion frequency of ~1 × 10^−7^ was far below that observed in *S. pombe* cells at the non-quiescent stage [56,78] and somewhat similar to that observed in cultured human cells [45]. In contrast to their MMEJ dependency in human cells, telomere fusions in quiescent *S. cerevisiae* cells showed a dependency on the NHEJ mechanism. Consistently, chromosome ends involved in the fusion typically contained long telomere repeats, suggesting the spontaneous occurrence of telomere-erosion-independent telomere inactivation, followed by NHEJ-mediated fusion [61]. The reduced frequency and NHEJ dependency of telomere fusions observed in the *S. cerevisiae* centromere deletion assay were consistent with the early investigation of spontaneous telomere fusions in wild-type *S. cerevisiae* using quantitative PCR [38].

## 3. Centromere Inactivation: A Manifestation of Functional Centromere Detachment

Unlike telomeres, which are inherently vulnerable to erosion, centromeres essentially exhibit no loss or deletion potential, making the analysis of spontaneous centromere detachment (Figure 4) more challenging. Most of the centromere losses have been observed only in mutants defective in the epigenetic inheritance and stability of CENP-A and other centromere-binding proteins [8,9]. However, spontaneous occurrence of centromere inactivation was reported in otherwise unstable dicentric chromosomes in clinical cases and model organisms [10,11]. Given the certainty of spontaneous telomere fusion (as discussed above), centromere inactivation can be considered as a functional centromere detachment event that occurs either before or after telomere fusion. The identification of native telomere fusion in the centromere deletion assay in *S. pombe* [77] and *S. cerevisiae* [61] is said to be an artificial recapitulation of the scenario of centromere inactivation followed by telomere fusion. Nevertheless, direct capture of spontaneous centromere inactivation remains a difficult task, and special experimental setups have been prepared to tackle this issue.

### 3.1. Centromere Inactivation in the Induced Human Dicentric Chromosomes

To study spontaneous centromere inactivation, an elegant attempt was made using cultured human cells by inducing transient inactivation of the telomere capping protein TRF2. Five of the twenty-three pairs of human chromosomes are acrocentric; i.e., their short arms consist entirely of repetitive units, each of which represents a ribosomal RNA gene (rDNA), and are highly condensed. In cultured human cells, transient TRF2 inactivation deprotected the telomeres preferentially in the short arms of these five acrocentric chromosomes, prompting their fusion [53]. The inter-centromere distance of the resulting dicentric chromosomes varied because of the differential DSBs in the short arms of acrocentric chromosomes prior to fusion [84]. However, instead of initiating the BFB cycle or cellular senescence, as observed under complete TRF2 deficiency [36], these dicentric chromosomes were generally stable and were accurately inherited across cell divisions, possibly because of the close proximity of the two centromeres [53].

Furthermore, during continuous cell cultures, inactivation of one of the two centromeres was observed in up to 50% of the dicentric chromosomes with larger inter-centromere distance. Centromere inactivation was associated with a significant reduction in the number of α-satellite repeats, suggesting the occurrence of an internal deletion within the centromeric DNA [53] (Table 1). A similar deletion-mediated centromere inactivation was observed in stabilized dicentric chromosomes in clinical cases [85,86,87], but this was valuable in that its spontaneous occurrence was observed in real-time in the laboratory.

### 3.2. Centromere Inactivation in the Induced Dicentric Chromosomes in S. pombe

Another fine experimental demonstration of spontaneous centromere inactivation was given in *S. pombe* [57]. The basic principle employed in this study was common to that used in the centromere deletion assay [77] (Figure 3), and two different approaches were undertaken for the artificial induction of dicentric chromosomes. One approach was an engineered site-specific recombination between the subtelomeric regions of different chromosomes, resulting in the tandem fusion of chromosomes and the simultaneous creation of a new selectable marker gene at the recombination site (Figure 3). The second approach was a natural meiotic crossing between wild-type and fused chromosome-containing cells obtained previously in the centromere deletion assay [77], followed by the genetic selection of the meiotic segregants harboring fused chromosomes in which the centromere-deleted segment was exchanged with the wild-type centromere through recombination [57].

Although these two approaches were completely independent, both yielded similar results, where the majority of cells with dicentric chromosomes died, while a set of cells classified into three types were produced. One type of the surviving *S. pombe* cells, obtained at an average frequency of ~7 × 10^−4^, exhibited a large deletion in either of the two centromeric regions of dicentric chromosomes [57] (Table 1). This is similar to the mechanism of centromere inactivation reported in the acrocentric chromosome-derived dicentric chromosomes in human cells, despite the low structural conservation of the centromere [53]. The frequency of centromere inactivation also differed greatly between *S. pombe* and human cells, which may partly reflect that the cells can tolerate dicentric chromosomes. The second type of *S. pombe* survivors, obtained at a frequency of ~8 × 10^−4^, showed a break at a site between the two functional centromeres of dicentric chromosomes [57] (Table 1). The occurrence of such chromosome breaks was consistent with that predicted during the BFB cycle (Figure 1). These broken ends were further stabilized in the isolated survivors by the de novo addition of telomeres, a phenomenon known as telomere healing (see Section 4), which additionally represented the dynamic behavior of chromosomes during rearrangements (Figure 1). The third type of *S. pombe* survivors, which appeared at a frequency of ~3 × 10^−3^ and accounted for two-thirds of all survivors, exhibited intact DNA at both centromeric regions of dicentric chromosomes but showed complete dissociation of CENP-A from either centromeric region [57] (Table 1). Inactivation of centromeres in the dicentric chromosomes through such epigenetic mechanisms had already been noted in the analyses of human cells in clinical cases [86,88], but an analysis in *S. pombe* nicely demonstrated the spontaneous occurrence of centromere inactivation after dicentric chromosome formation and also indicated its efficiency relative to that of other responses. Additionally, heterochromatin formation was reported along the epigenetically inactivated centromeres in *S. pombe* survivors, reminiscent of the dicentric chromosomes in human cells [89], but heterochromatinization here served only to maintain the inactive state of centromeres, not to trigger the inactivation [57].

### 3.3. Centromere Inactivation in the Induced Dicentric Chromosomes in Various Yeast Species

Spontaneous centromere inactivation was also demonstrated in the evolutionarily distant basidiomycete yeast *Cryptococcus deuterogattii*, a human pathogen, using a similar artificial induction of dicentric chromosomes [68]. Instead of inducing site-specific recombination, CRISPR-Cas9-mediated genome editing technology was employed in *C. deuterogattii* to fuse two chromosome-ends together with a selectable marker gene in between (Figure 3). Genetic selection led to the identification of two isolates of chromosome fusion transformants harboring either a break in the fused chromosome at the inter-centromeric site or an epigenetically inactivated centromere [68] (Table 1). The transient nature of the CRISPR-Cas9 treatment made it difficult to compare the efficiency of events in *C. deuterogattii* with those in *S. pombe*, but the responses to dicentric chromosome formation appear to have been conserved during evolution. Interestingly, the CENP-A accumulation pattern in the epigenetically stabilized dicentric chromosome suggested that both centromeres may be alternately inactivated during cell culture, as proposed previously in *Drosophila* strains [90].

Studies in *S. cerevisiae* also provided a good demonstration of the fate of dicentric chromosomes. The functional centromere in *S. cerevisiae*, whose DNA was the first to be cloned and sequenced among eukaryotes, is exceptional, given that it is genetically defined by a unique 125-bp DNA sequence rather than by the presence of CENP-A [3]. Therefore, epigenetic centromere inactivation has never been observed as a secondary response to dicentric chromosome induction. Instead, the breakage of the fused chromosome, which serves as a useful target for the analysis of DSB healing (see Section 4), and the local deletion of centromeric DNA account for all the responses to dicentric chromosomes, as demonstrated through various means of dicentric chromosome induction, including meiotic recombination [91,92], ectopic integration of centromeric DNA [62,63] (Table 1), and genetic selection of spontaneous rearrangements [93]. Functional inactivation of the centromere by forced transcription of centromeric DNA (Figure 3) is also a powerful means of conditionally inducing dicentric chromosomes [81,83,94,95] and nicely clarifies the repair pathways against subsequent chromosome breaks [96].

## 4. Telomere Healing: A Manifestation of Functional Telomere Attachment

Considering the mode of action of telomerase on native telomeric DNA, it is relatively easy to envision telomerase-mediated de novo telomere addition (telomere healing) to non-telomeric DSB ends (Figure 5) as a manifestation of flexible telomere attachment. Such a reaction was indeed found to occur spontaneously at the broken ends of dicentric chromosomes in *S. pombe*, as described in Section 3 [57] (Figure 1). Moreover, telomere healing contributed to the in vivo creation of centromeric minichromosomes in *S. pombe* [97] by stabilizing the DSB ends induced by γ-irradiation, leading to the elimination of entire chromosome arms from authentic chromosomes [98]. Through the same mechanism, telomere repeat sequences at the ends of a linearized plasmid have long been shown to seed the formation of new telomeres when reintroduced into *S. cerevisiae* [1,99] and *S. pombe* [74], the latter being used further in the minichromosome circularization assay described above [56]. Based on these observations, it could be speculated that telomere healing is easily possible in vivo, but the presence of DSBs is a definite prerequisite. Irrespective of the efficiency of telomere healing, careful consideration should be given to DSBs, especially when telomere healing is discussed in the native context.

### 4.1. Telomere Healing at Site-Specific DSBs in S. cerevisiae

When a single DSB was induced at a chromosome terminus harboring a counter-selectable marker gene in *S. cerevisiae* using a site-specific endonuclease, telomere healing was detected at the DSB end of the terminally truncated chromosome at a frequency lower than 1 × 10^−3^ in the wild-type context [60,100] (Table 1). HR-mediated genome maintenance mechanisms stabilized most of the DSB ends, whereas telomere healing represented only a small portion of the stabilization events, even in the *rad52* mutant background where HR should not be possible [60,100]. Therefore, contrary to the above expectation, telomere healing is a rare event in *S. cerevisiae*. The poor success of telomere healing could partly be attributed to the inhibition of the telomerase activity of Pif1 DNA helicase [60,101]. However, many other mechanisms including NHEJ and break-induced replication (BIR) [48,49] also acted redundantly on the DSB ends and overwhelmed the process of telomere healing [60,102,103], giving the impression that telomere healing is the last option among these radical genome maintenance mechanisms.

Analysis of endogenous DNA sequences contributing to telomere healing suggested sequence preference at the junction as well as surrounding DNA for successful telomere repeat addition [60,100]. Furthermore, introduction of short tracts of telomere repeats into the DNA near the DSB site greatly restored telomere healing to >90% efficiency in the wild type [64] (Table 1). The linkage between telomere repeats and endogenous DNA sequences flanking the DSB site is a major factor affecting the regulation of telomere healing at site-specific DSBs [104,105,106,107]. This DNA sequence-dependent attachment of functional telomeres is a relatively unique feature among the other flexible characteristics of the detachment and attachment of centromeres and telomeres.

### 4.2. Telomere Healing during GCRs in S. cerevisiae

The above studies of site-specific DSB-mediated rearrangements indicated that telomere healing is not a random event. This was independently confirmed in *S. cerevisiae* using the non-biased GCR assay [108], which selects for the spontaneous loss of two counter-selectable marker genes located adjacent to each other on the terminal non-essential region of a chromosome and allows the isolation of various chromosomal rearrangements including telomere healing [65,109]. Given the low probability of the simultaneous occurrence of loss-of-function mutations in the two marker genes (~2 × 10^−13^, an estimate based on the rate of a single mutation [65]), the non-biased GCR assay shows high specificity and high sensitivity for the GCR events. In addition, all the events included in GCRs, ranging from the initial DSB to secondary chromosomal rearrangements after the BFB-like cycle, occur spontaneously in this assay, allowing the dissection of molecular mechanisms using genetic approaches [18].

In wild-type GCRs, telomere healing was observed at the broken ends at a frequency of ~3 × 10^−10^ [65,66] (Table 1). Sequence analysis of over 500 breakpoints, which were scattered along the 12-kb region between the marker gene and the first essential gene on the centromere-proximal side of the chromosome, revealed that these breakpoints were not distributed at random but were localized at certain points, forming hotspots; moreover, a bias against the centromeric side and a clear preference for the telomere repeat-matching sequence at the junction were observed [108]. Such telomere healing events accounted for more than 80% of the wild-type GCRs [65], which is in contrast to their minor contribution to the stabilization of site-specific DSBs [60,100]. Consistently, when the site-specific DSB induction was combined with the GCR assay, the contribution of telomere healing to GCRs was masked by other overwhelming mechanisms such as HR, NHEJ, MMEJ, and BIR [110]. However, combination of the GCR assay with factors causing global DNA damage, such as γ-irradiation and DNA alkylating agents, led to different results; these DNA-damaging agents increased GCRs in a dose-dependent manner, and telomere healing accounted for most of the GCRs unless base substitution mutations were stimulated [110]. Telomere healing may be activated by specific types of DSBs but not by those introduced by endonucleases. In this regard, the DSBs for GCRs in the native (wild type) context are assumed to have resulted from DNA replication errors that occurred when replication folks encounter difficult-to-copy templates [18,66,111]. GCRs may be inherently associated with eukaryotic chromosomes, along which multiple replication origins are scattered randomly.

### 4.3. Evolutionary Conservation of Telomere Healing during GCRs

The GCR assay was recently established in *S. pombe*, which estimated the frequency of telomere healing in the wild-type context as ~2 × 10^−9^ and demonstrated its evolutionary conservation as the primal GCR event [59] (Table 1). The rate of GCRs per unit length of DNA was matched fairly well between *S. cerevisiae* (~3 × 10^−11^/kb) [109] and *S. pombe* (~1 × 10^−10^/kb) [59], suggesting a universal, DNA sequence-independent, and DNA length-dependent mechanism for the development of DSBs as a GCR trigger.

A similar non-biased selection for chromosomal rearrangements upon spontaneous loss of a telomeric region was also performed in a telomerase-positive human cancer cell line using a counter-selectable marker gene, which identified telomere healing events at a frequency of ~1 × 10^−7^ [54] (Table 1). However, inter-sister chromatid fusions were yielded more frequently in the same assay, and telomere healing represented only 11% of all GCR events [54], which was further decreased to 4% when site-specific DSBs were induced in the GCR assay [112]. It remains unclear whether the differences in telomere healing results between yeast species and humans arise from interspecies variation or cancer cell-specific features.

In the GCR assay in *S. pombe*, the telomere capping proteins Taz1 and Rap1 suppressed telomere healing [59], which is somewhat reminiscent of telomere healing regulation at the site-specific DSB in *S. cerevisiae* [104,106]. However, the regulation in *S. cerevisiae* was fully dependent on the telomeric repeat sequences near the DSB, whereas no preference for telomere repeat-like sequences in the DNA surrounding various breakpoints was found in the GCR assay in both *S. cerevisiae* and *S. pombe* [59,108]. Instead, direct involvement of *S. pombe* Taz1 and Rap1 proteins in the DSB repair pathway was proposed based on a different assay using site-specific endonuclease induction [59]. It is also possible that the mechanism of telomerase recruitment to the endonuclease-induced site-specific DSBs, leading to telomere healing, is not conserved between *S. cerevisiae* and *S. pombe*.

### 4.4. Site-Specific DSBs, GCRs, and Telomere Healing in Centromeric Minichromosomes in S. pombe

Instead of examining the effect of site-specific DSBs on chromosomal GCRs, the outcomes of site-specific DSBs with respect to telomere healing were explored in *S. pombe* using a different assay. The centromeric minichromosomes [97] were used as targets for site-specific DSBs in this assay [113], and the fate of terminally truncated minichromosomes was monitored using specific genetic markers [114]. Reminiscent of the site-specific DSB lacking nearby telomere repeats in *S. cerevisiae* [60,100], complex HR-mediated mechanisms dominated over telomere healing for repairing the broken DNA ends [114,115,116]. Therefore, it seems likely that at least the priority of repair mechanisms is evolutionarily conserved. It was proposed recently that telomere healing could occur as an aberrant byproduct of the HR intermediates [117]. Nonetheless, telomere healing seems to be the last option for genome rearrangements.

Intriguingly, the preference of centromeric minichromosomes over HR-mediated rearrangements was also observed in the GCR assay without site-specific DSBs [118]. The centromeric region of the minichromosome was subjected to several HR reactions in this spontaneous GCR assay, and telomere healing was detected only when these HR pathways were inactivated by various mutations [118,119,120]. The HR-mediated GCRs were shown to be under the control of DNA replication [119]. It is interesting to consider the difference in *S. pombe* GCR assay results between authentic chromosomes and minichromosomes, given the involvement of DNA replication.

## 5. Neocentromere Formation: A Manifestation of Functional Centromere Attachment

Successful telomere healing in the native condition results in an increase in chromosome number by one. However, this increase in chromosome number is not effective until centromere function is acquired somewhere along the length of the chromosome (Figure 1). The formation of neocentromeres, i.e., ectopic centromeres established at non-centromeric regions, is a perfect example of centromere function acquisition (Figure 6). Neocentromeres are suitable for elucidating the epigenetic characters of centromeres and have been studied extensively since their first discovery in human clinical cases [34,121]. However, the estimation of neocentromere formation frequency under normal conditions remains challenging. Spontaneous acquisition of neocentromeres destabilizes normal chromosomes, and the stable detection of neocentromeres requires their association with secondary rearrangements within the chromosome (Figure 1). While this kind of difficulty may be common to the other chromosomal events discussed so far, neocentromere formation is particularly difficult to study, since it is an epigenetic event and has no prerequisite, such as DSBs in the case of telomere healing. The centromere deletion assay, as mentioned earlier in Section 2 (Figure 3), provides a solution to this problem. Centromere deletion artificially induces the formation of acentric chromosomes, and neocentromere formation can be detected as a rescue event for the acentric chromosome, in addition to telomere fusion if the frequency of both is within the same range.

### 5.1. Efficiency of Neocentromere Formation in Addition to Telomere Fusion in Various Yeast Species

The first attempt to induce centromere deletion was made in *S. pombe* using a site-specific recombinase (Figure 3), which led to the detection of both neocentromere formation and telomere fusion in the surviving cells [77]. The average frequency of neocentromere formation was estimated as ~2 × 10^−4^ (Table 1), which was within the same range as the telomere fusion frequency [58,77]. Similarly, in *S. cerevisiae*, a centromere deletion assay yielded only telomere fusion survivors at a frequency of ~1 × 10^−7^ [61], suggesting that neocentromere formation is highly infrequent in *S. cerevisiae*. This result reinforces the results of the centromere inactivation assay involving forced transcription of centromeric DNA [94], and confirms that centromeres in *S. cerevisiae* are genetically determined, although de novo centromere formation on the centromeric DNA is at least partly epigenetically controlled in this organism [122]. On the contrary, neocentromere formation occurred highly efficiently in the human commensal yeast *Candida albicans* [69,70]. Centromere deletion was performed by conventional genome replacement, i.e., the introduction of a linear DNA fragment showing homology to the sequence flanking the target centromeric region, and the deletion transformants were obtained with nearly 100% efficiency, all of them harboring neocentromeres in the centromere-deleted chromosomes (Table 1). This observation, however, does not rule out the possibility that spontaneous telomere fusions occur in *C. albicans*, in addition to the frequent neocentromere formation, at a rate similar to or greater than that observed in *S. cerevisiae* [61,123]. Efficient neocentromere formation appears to be common to *Candida* species [70] and independent of the centromere deletion methods, because the same transformation-based conventional centromere deletion in another pathogenic yeast, *C. deuterogattii*, yielded neocentromere formation at a frequency lower than 1 × 10^−3^ [67] (Table 1).

### 5.2. Efficiency of Neocentromere Formation in Addition to Telomere Fusion in Vertebrate Cells

The centromere deletion assay was also performed in vertebrate cells (Figure 3). An elaborate site-specific recombination-mediated centromere deletion, combined with genetic selection, in chicken DT40 cells led to the isolation of neocentromere-forming cells at a frequency of ~3 × 10^−6^ [71] (Table 1). A telomere fusion event was also observed in the same screen, at a frequency of ~3 × 10^−7^ [71] (Table 1), which is less than the spontaneous telomere fusion frequency observed in *S. pombe* but within the range of that observed in *S. cerevisiae* and cultured human cells (see Section 2) [45,58,61].

In addition, a similar centromere deletion assay was also recently performed in cultured human cells [55]. In this case, centromere deletion was designed to take the centromere out of the chromosome as a circular DNA similar to the site-specific recombination-mediated method; however, CRISPR-Cas9 was used to induce DSBs, and centromeric DNA circularization was monitored by the restoration of the expression of a fluorescent marker gene originally located at the DSB ends, reminiscent of the minichromosome circularization assay in *S. pombe* (see Section 2) (Figure 3) [56]. After considerable cell sorting based on fluorescent signals, neocentromere-forming human cells were finally obtained at a frequency of ~8 × 10^−6^ [55] (Table 1). Other than neocentromere formation, tetraploidization was observed as a cellular response to centromere deletion; however, telomere fusion was not detected, which is consistent with the estimated frequency of spontaneous telomere fusion in human cells [45].

### 5.3. Site Preference for Neocentromere Formation

The centromere deletion assay revealed that neocentromeres do not form randomly along the chromosomes, despite their epigenetic nature. The ability to provide a suitable chromosomal environment for neocentromere formation may affect the measured efficiency of neocentromere formation in each organism. *S. pombe* neocentromeres, the efficiency of which was measured to be relatively high, were found almost exclusively at the subtelomeric region adjacent to heterochromatin [58,77]. Similar site preference was also observed in the analysis of centromere reactivation in dicentric *S. pombe* chromosomes and in the investigation of centromeric activity of ectopic DNAs at authentic *S. pombe* loci [57,124], suggesting that the site preference for neocentromere formation is intrinsic to *S. pombe*. Although subtelomeric heterochromatin was involved in neocentromere formation, it was not the primary determinant of neocentromere sites [58]. An unknown mechanism associated with subtelomeric regions seems to confer preferential environment for neocentromere formation in *S. pombe*.

Similar telomeric neocentromere formation was also observed in two pathogenic yeast species, *C. albicans* and *C. deuterogattii*, and chicken DT40 cells [67,69,71]. However, in these organisms, each telomeric neocentromere represented only a minor fraction of all identified neocentromeres, unlike that in *S. pombe*. Human neocentromere was also non-telomeric [55]. In yeast species, chicken DT40 cells, and human cultured cells, the most popular feature of neocentromeres was their formation at sites located near the original centromere [55,67,69,70,71]. These sites were not directly adjacent to the original centromere but were in close proximity. Additionally, these sites were not associated with heterochromatin; instead, they corresponded either to actively transcribed regions or to silent or intergenic regions. Similar neocentromere formation proximal to the original centromere was also reported in an early study on *Drosophila* [125]. Although the actual mechanisms that promote the formation of neocentromeres remain to be elucidated, a three-dimensional nuclear structure consisting of multiple chromosomes has been proposed to play an important role in their regulation [70,126,127,128,129].

### 5.4. Analysis of Ongoing Chromosomal Rearrangements and Maturation

Neocentromere formation efficiency determined via the centromere deletion assay is specific to the chromosome from which the centromere is deleted and will not necessarily be the representative of an organism. For example, in *S. pombe*, neocentromere formation efficiency observed in chromosome III was two orders of magnitude lower than that observed in other chromosomes [58]. This is because subtelomeric regions preferred for neocentromere formation were occupied by rDNA repeats in chromosome III. Therefore, the resultant neocentromeres formed infrequently within the rDNA repeats or in the subtelomere-distal region were unstable. Moreover, the spontaneous loss of these neocentromeres and subsequent telomere fusions, which connected the neocentromere-lacking acentric chromosome III with another chromosome, appeared recurrently during cell culture [58]. Conversely, maturation from unstable to stable neocentromeres via genetic and epigenetic alterations was observed in chromosome III during continuous cell culture [58]. Similar culture passage-dependent maturation of neocentromeres was also observed in cultured human cells [55]. Furthermore, in *C. deuterogattii*, telomere fusion could not be detected in the centromere deletion assay, even with the use of the efficient CRISPR-Cas9-mediated centromere deletion method (Figure 3), but was frequently detected in the successive cultures of neocentromere-forming cells at an elevated temperature, which seemed to cause stochastic neocentromere loss from the chromosome [67] (Table 1). These series of spontaneous rearrangement events initiated by centromere deletion still have much to offer and are promising targets for elucidating spontaneous chromosomal alterations ongoing in the laboratory.

## 6. Overall Discussion and Future Perspectives

In this review, we discussed various topics related to the chromosomal detachment and attachment of centromeres and telomeres, with an emphasis on their frequency. Most of the events are very rare in the wild-type context, which highlights the high-fidelity maintenance of multiple chromosomes in the nucleus. To capture these rare events, elaborate experimental setups have been developed (Figure 3). Interestingly, the scheme of an experimental setup aiming to detect one event shows similarities to the target event of another experimental setup, such as the centromere deletion assay and centromere inactivation and the dicentric chromosome induction assay and telomere fusion. These similarities raise the possibility that individual events discussed separately in this review are indeed closely linked as intranuclear cause-and-effect relationships (Figure 1). Thus, the final frequency of sequential events could be calculated by multiplying the individual frequencies of all events involved. Alternatively, the frequency of an event may represent that event only as an initial trigger and may change when that event is considered as one of the secondary rearrangements. The GCR assay in *S. cerevisiae* and *S. pombe* may possibly provide a clue to this issue [59,65,66,93]. According to this point of view, it will be intriguing to examine the frequencies of sequential rearrangements that occur spontaneously and recurrently during continuous cell culture [53,57,58,67].

We are now entering a new era of synthetic biology. The CRISPR-Cas9-mediated centromere deletion assay and dicentric chromosome induction (Figure 3) were applied simultaneously and repeatedly to the chromosomes of *S. cerevisiae*, leading to the creation of strains harboring two linear chromosomes [130], a single linear chromosome [131], or a single circular chromosome (as in bacteria) [132]. A single linear chromosome-harboring *S. pombe* strain was also generated recently using a similar approach [133]. These approaches used the same techniques as described in this review but with a different intention. The above approaches induced all chromosomal outcomes artificially and left no chance for a spontaneous response, whereas the studies described in this review allowed the cells to generate chromosomal rearrangements naturally after the initial designed trigger. Both approaches have their own advantages. New division of labor was revealed among the *S. cerevisiae* telomere capping proteins only by analyzing the telomere fusion event in the single chromosome yeast strain, similar to the minichromosome circularization study in *S. pombe* [56,134]. The former approach may lead to additional surprise discoveries, but serendipity is expected almost exclusively from the latter approach. In conclusion, different means should be used for different purposes.

Origins of replication play a crucial role in cell tolerance of spontaneous chromosomal rearrangements. The flexible chromosome configuration changes are made possible only by the multiple replication origins present redundantly on the chromosomes. If a chromosome possessed only one tightly regulated origin of replication, with no chance for dormant origins along the chromosome, many possible chromosomal rearrangements would be compromised and never work successfully. Given the leading role of DNA replication in the GCR reaction [66,111,119], eukaryotic DNA replication is important not only for tolerating the DNA rearrangements but also for promoting them. Even though complete detachment of origins of replication from the chromosomes is impossible and therefore should not be considered, it would be fascinating to consider the remaining detachment and attachment events in combination with DNA replication, which is central to the function of the nucleus.

## Figures and Tables

**Figure 1 biomolecules-13-01016-f001:**
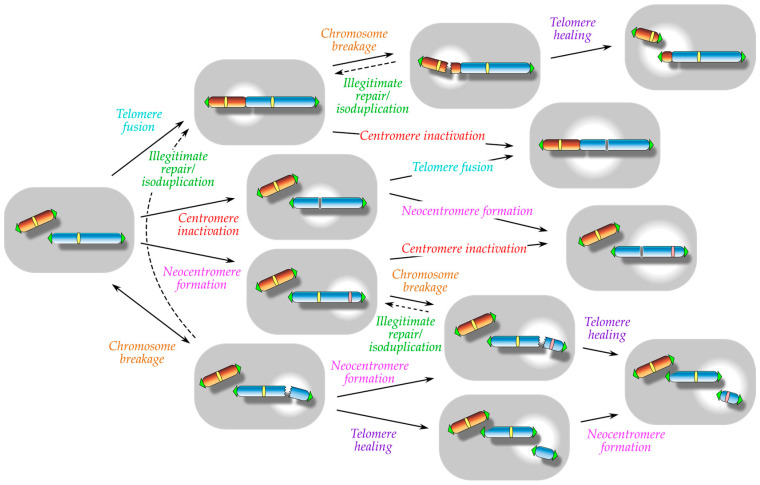
Chromosomal alterations typically associated with spontaneous detachment and attachment of telomeres and centromeres. Type of alteration is distinguished by color. The same event may work catastrophically or beneficially depending on the chromosomal conditions. The same outcome may be reached from the events in different orders. When a chromosome break occurs (chromosome breakage), illegitimate DSB repair or inter-sister chromatid fusion (isoduplication) may subsequently generate a new dicentric chromosome leading to a breakage–fusion–bridge cycle [21], which is not drawn accurately and instead is expressed crudely by an arrow with a dashed line.

**Figure 2 biomolecules-13-01016-f002:**
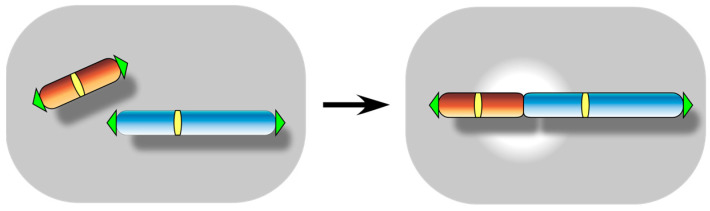
Telomere fusion event. It is also referred to as chromosome end-to-end fusion or telomere-to-telomere fusion. It occurs essentially when chromosome ends are recognized by cellular DSB repair pathways as DNA damages.

**Figure 3 biomolecules-13-01016-f003:**
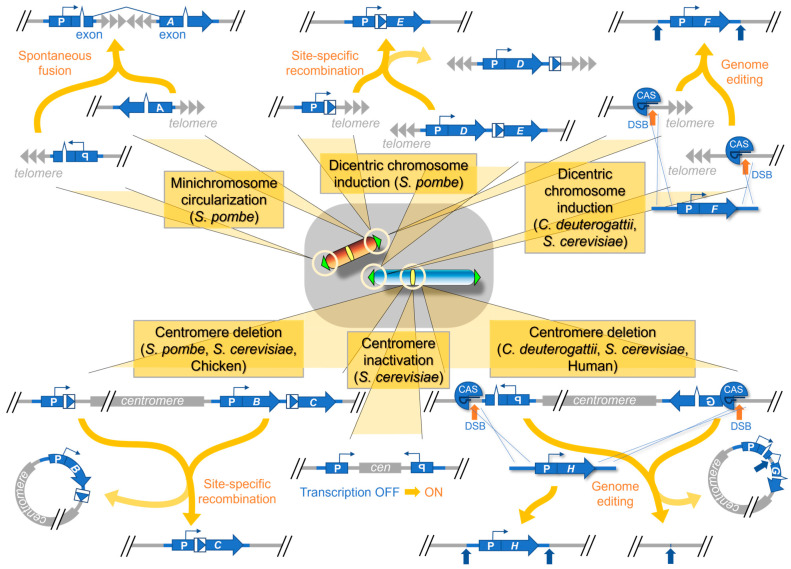
Representative experimental setups to capture spontaneous chromosomal alteration events. In every setup, coding regions of selectable, counter-selectable, or fluorescent marker genes (*A*–*G*) are used appropriately in combination with appropriate promoters (P) in an appropriate arrangement to monitor the desired rearrangement by changes in gene expression. Essentially, the minichromosome circularization assay in *S. pombe* utilizes RNA splicing, the centromere deletion assay in *S. pombe*, *S. cerevisiae* or chicken and the dicentric chromosome induction in *S. pombe* utilize site-specific recombination, the centromere inactivation assay in *S. cerevisiae* utilizes forced transcription, and the dicentric chromosome induction assay in *C. deuterogattii* or *S cerevisiae* and the centromere deletion assay in *C. deuterogattii*, *S. cerevisiae* or human cells utilize CRISPR-Cas9-mediated genome editing for selection. Byproducts are sometimes generated and are indicated by light colored arrows. The same assays performed in different organisms are shown together. See text for details.

**Figure 4 biomolecules-13-01016-f004:**
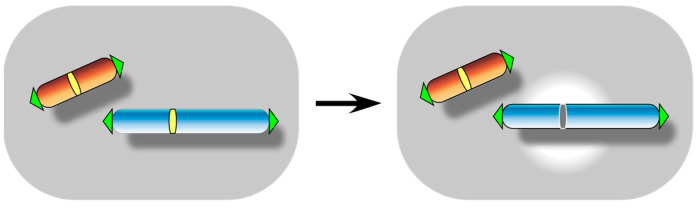
Centromere inactivation event. It results in the formation of a dysfunctional acentric chromosome when it occurs to a monocentric chromosome.

**Figure 5 biomolecules-13-01016-f005:**
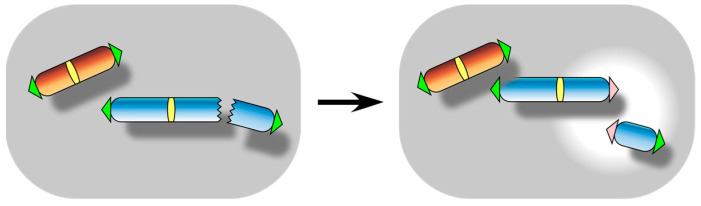
Telomere healing event. It is also referred to as chromosome healing or de novo telomere addition. It essentially stabilizes the broken DNA ends and protects them from DSB repair mechanisms.

**Figure 6 biomolecules-13-01016-f006:**
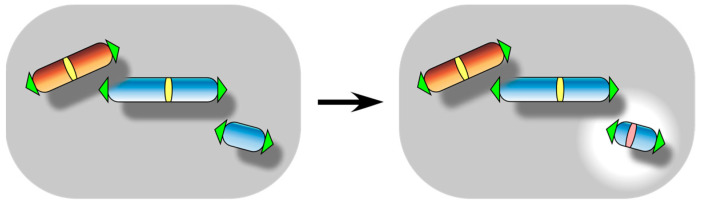
Neocentromere formation event. It occurs to non-centromeric DNA regions epigenetically by stable incorporation and propagation of centromere-specific histone H3 variant CENP-A.

**Table 1 biomolecules-13-01016-t001:** Experimentally estimated frequencies of chromosomal events per given chromosome per cell in the wild-type context.

	Telomere Fusion	Centromere Inactivation		Telomere Healing	Neocentromere Formation
		(DNA Deletion)	(Epigenetic Loss)		
Human *^1^	~4 × 10^−6^ [45]	<50% [53]	—	~1 × 10^−7^ [54]	~8 × 10^−6^ [55]
*S. pombe*	~1 × 10^−4^ *^2^ [56]	~7 × 10^−4^ [57]	~3 × 10^−3^ [57]	~8 × 10^−4^ *^5^ [57]	~2 × 10^−4^ [58]
	~6 × 10^−5^ *^3^ [58]			~2 × 10^−9^ *^6^ [59]	
*S. cerevisiae*				<1 × 10^−3^ *^7^ [60]	
	~1 × 10^−7^ [61]	~2% [62,63]	—	>90% *^8^ [64]	<1 × 10^−7^ [61]
				~3 × 10^−10^ *^6^ [65,66]	
*C. deuterogattii*	*u.f.* *^4^ [67]	—	*u.f.* *^4^ [68]	—	<1 × 10^−3^ [67]
*C. albicans*	—	—	—	—	~100% [69,70]
Chicken	~3 × 10^−7^ [71]	—	—	—	~3 × 10^−6^ [71]

*^1^ including results in different cell lines; *^2^ in minichromosome circularization assay; *^3^ in centromere deletion assay; *^4^ experimentally detected with unknown frequency; *^5^ in dicentric chromosome induction assay; *^6^ in GCR assay; *^7^ in site-specific DSB induction assay; *^8^ in telomeric repeat-associated site-specific DSB induction assay.

## Data Availability

Not applicable.

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
