# Peer review of "Flexible Attachment and Detachment of Centromeres and Telomeres to and from Chromosomes"

_biomolecules, 2023, doi:10.3390/biom13061016_

Round 1

Reviewer 1 Report

This reviews summarizes a large body of work on various aspects of telomere and centromere rearrangements, fusions, deletions and inactivation in a number of organisms. As stated by the authors, the review does "not dive into the mechanistic details". The consequence of this is that the review is very phenomenological. We learn about various events and the frequency of those events, but their is little if any mechanistic insight.

My concern is in regard to the target audience. The review is not useful for aficionados. It is also not particularly useful for novices, as that audience will not understand the methods, unless they are very familiar with the organism and how to work with it. The review may be useful for graduate students who are working in the field and need to broaden their knowledge base to other organisms.

Specific comments:

The authors state that dicentric chromosomes are "torn" apart in mitosis (line 120). This is unlikely to be accurate. Microtubules and kinetochores generate force in the 1-10 piconewton range, while the force required to break covalent bonds holding DNA together are in the hundreds of piconewton range. A much more likely mechanism of breakage is enzymatic, regression of a replication force, or severing the DNA following cytokinesis. While I understand the authors rationale to keep it simple, "torn" is a poor descriptor of the actual process.

The authors give a very short description about the fate of the second centromere following dicentric breakage in budding yeast (lines 308-321). This is the system we know the most about. The authors state that "local deletion of centromeric DNA account for all the responses to dicentric chromosomes" (line 315-316). This is not correct. Depending on where the second centromere is in the chromosome, the responses include homologous recombination (that yields a ring and rod monocentric derivative) or NHEJ (see Cook et al., PLoS Genet. 2021 Mar; 17(3): e1009442. Published online 2021 Mar 18. doi: 10.1371/journal.pgen.1009442).

The authors state that neocentromere formation is highly infrequent in budding yeast. While this is true, there are reports that distinguish between de novo centromere formation vs. template-directed centromere formation (J Cell Biol. 2003 Mar 17; 160(6): 833–843. doi: 10.1083/jcb.200211116 PMCID:PMC2173759). In an organism such as budding yeast, it is likely that the specific requirements for de novo centromere formation are analogous to neocentromere formation, both of which require the building of centromere without a pre-existing template.

Small comments: I think the authors mean "reduced frequency and NHEJ" rather than "deduced frequency and NHEJ" (line 214).

,

The language is OK, but there are numerous places that would benefit from a careful edit.

Author Response

Below, brown text indicates the reviewer's original comments, black text indicates the point-by-point responses by the authors, and black italicized text indicates the revisions made in the revised manuscript, with actual revisions from the previous manuscript highlighted in gray.

This reviews summarizes a large body of work on various aspects of telomere and centromere rearrangements, fusions, deletions and inactivation in a number of organisms. As stated by the authors, the review does "not dive into the mechanistic details". The consequence of this is that the review is very phenomenological. We learn about various events and the frequency of those events, but their is little if any mechanistic insight.

My concern is in regard to the target audience. The review is not useful for aficionados. It is also not particularly useful for novices, as that audience will not understand the methods, unless they are very familiar with the organism and how to work with it. The review may be useful for graduate students who are working in the field and need to broaden their knowledge base to other organisms.

We thank the reviewer for his/her close reading and insightful comments on our manuscript. We agree with the reviewer that this review is largely phenomenological, providing few mechanistic insights. We understand the reviewer’s worry about the scope of readership. However, a wide variety of chromosomal events are introduced and discussed together in this review. To our knowledge, there has been no other article covering the broad events equally like this. Exposing different events under the same criterion, i.e., in wild type and not in mutants involved in one particular event, simultaneously in a single article may help the readers to gain a broaden and integrated perspective on the spontaneous chromosomal reorganization. All the events occur naturally in the same background and they may be interconnected. Our previous manuscript mentioned this point poorly, and we have corrected the text and have cited more articles in the introduction section of the revised manuscript as follows:

Lines 82-89

In addition, this review does not dive into the mechanistic details of each event but introduces only the bare minimum of what is known. For detailed mechanisms, readers are directed to many excellent reviews on each topic [13,18,26-35]. Instead, this review sheds light on the natural frequencies of events in the wild type. This allows a comparison of the efficiencies of different events in native context and permits combining these efficiencies in the future, which may be useful in obtaining an integrated perspective on chromosome flexibility (Figure 1).

Specific comments:

The authors state that dicentric chromosomes are "torn" apart in mitosis (line 120). This is unlikely to be accurate. Microtubules and kinetochores generate force in the 1-10 piconewton range, while the force required to break covalent bonds holding DNA together are in the hundreds of piconewton range. A much more likely mechanism of breakage is enzymatic, regression of a replication force, or severing the DNA following cytokinesis. While I understand the authors rationale to keep it simple, "torn" is a poor descriptor of the actual process.

We thank the reviewers for this sharp comment. Our explanation for BFB cycle in previous manuscript was an ill-considered expression. We agree with the reviewer that it was inaccurate to explain the actual events in the cells. We have corrected the sentence in section 2.2 of the revised manuscript as follows:

Lines 138-142

Telomere fusion is often recognized as a part of consecutive and complex chromosomal rearrangements, because it results in the formation of a dicentric chromosome, which leads to chromosome breakage in the following cell division and initiates further rounds of broken-end fusion and dicentric chromosome formation, a phenomenon called breakage-fusion-bridge (BFB) cycle originally proposed by McClintock (Figure 1) [13,20,21,72].

The authors give a very short description about the fate of the second centromere following dicentric breakage in budding yeast (lines 308-321). This is the system we know the most about. The authors state that "local deletion of centromeric DNA account for all the responses to dicentric chromosomes" (line 315-316). This is not correct. Depending on where the second centromere is in the chromosome, the responses include homologous recombination (that yields a ring and rod monocentric derivative) or NHEJ (see Cook et al., PLoS Genet. 2021 Mar; 17(3): e1009442. Published online 2021 Mar 18. doi: 10.1371/journal.pgen.1009442).

The paragraph the reviewer pointed out is in section 3 and discusses particularly about the spontaneous event of centromere inactivation detected to happen after dicentric chromosome formation in S. cerevisiae. It is not on the general destinies of S. cerevisiae dicentric chromosomes. Although the reviewer’s point is correct, it falls out of the scope. We apologize for the confusion we made. We also agree that the work the reviewer raised is important. We have cited it and modified the sentence in the revised manuscript as follows:

Lines 356-359

Functional inactivation of the centromere by forced transcription of centromeric DNA (Figure 3) is also a powerful means of conditionally inducing dicentric chromosomes [81,83,94,95] and nicely clarifies the repair pathways against subsequent chromosome breaks [96].

The authors state that neocentromere formation is highly infrequent in budding yeast. While this is true, there are reports that distinguish between de novo centromere formation vs. template-directed centromere formation (J Cell Biol. 2003 Mar 17; 160(6): 833–843. doi: 10.1083/jcb.200211116 PMCID:PMC2173759). In an organism such as budding yeast, it is likely that the specific requirements for de novo centromere formation are analogous to neocentromere formation, both of which require the building of centromere without a pre-existing template.

Thank you for raising an important point which was missed in the previous manuscript. It should be mentioned in this review article. We have modified the description of neocentromere formation in S. cerevisiae in section 5.1 in the revised manuscript as follows:

Lines 517-521

This result reinforces the results of the centromere inactivation assay involving forced transcription of centromeric DNA [94], and confirms that centromeres in S. cerevisiae are genetically determined, although de novo centromere formation on the centromeric DNA is at least partly epigenetically controlled in this organism [122].

Small comments: I think the authors mean "reduced frequency and NHEJ" rather than "deduced frequency and NHEJ" (line 214).

Thank you for pointing this out. We have corrected it to “reduced” in the revised manuscript (line 246).

Reviewer 2 Report

In this review, the authors reviewed studies on the rearrangement of centromere and telomere sequences, focusing specifically on the frequency and methodology used to detect such rare events in wild-type cells. The article indeed provides a review of the information for the latter study. However, the authors tend to mix pathological/experimental findings with physiological findings. Furthermore, some wording and statements are vague and require further polishing to make the article clearer.

Here are the general comments:

1.          Some similar terms are provided without an explanation of the difference. For example, gross chromosomal rearrangements (GCRs)/chromosomal aberrations (CAs) in line 58 (especially only GCRs are mentioned in the later section), and reproductive isolation/speciation in line 72. It needs to be addressed whether these terms have the same meaning or if there are differences between them.

2.          In line 91, the authors state that "The special organization of telomeres is ensured by telomerase..." Telomerase is an enzyme that elongates telomere length but does not directly affect the structure of telomeres. Therefore, the relationship between the special organization of telomeres and telomerase needs to be described more clearly.

3.          In lines 94-95, the authors state that "telomeres are subject to uncapping and telomere fusions occur readily when the telomere capping protein or the telomerase enzyme is functionally compromised, as reported in mammals and various yeast species." This statement may be imprecise and need to modified since not all cells in mammals express telomerase. Does a cell with no telomerase expression have a higher frequency of telomere fusion?

4.          In this review, the authors emphasize the rare abnormal events that occur in wild-type cells. Yet, they focus on the frequency and methodology to detect them. The importance of these events must be highlighted in the article.

5.          In Figure 1, the authors depict different chromosomal events with arrowheads. Is there a specific order in which these events occur? Moreover, the definition of the different events is not provided in the figure or text, which needs to be described. It is recommended to describe different events in different figure panels.

6.          In Table 1, the authors list the frequency of different chromosome events in wild-type cells. However, the table is confusing. First, the unit or definition of frequency is not mentioned (e.g., per cell, per chromosome, or cells per day). Second, it seems like the table includes both the frequency of natural conditions and the frequency after artificial treatment (e.g., neocentromere formation in C. albicans). The separation of these two different conditions is required since they are not compatible. 

7.          In Figure 2, the authors represent different experimental setups. However, a detailed description of the design is lacking and needs to be added to the text. It is recommended to describe different experiments in different panels to make the figure clearer.

8.          Some methodologies are vague, such as quantitative PCR-based experiments to detect telomere fusions (Line 217) and non-biased GCR assay for telomere healing (Line 363). Since the article aims to review the methodology, the experimental design needs to be described clearly.

The article is understandable, however, some sentences may have imprecise wording or grammar error which could be further improved.

Author Response

Below, brown text indicates the reviewer's original comments, black text indicates the point-by-point responses by the authors, and black italicized text indicates the revisions made in the revised manuscript, with actual revisions from the previous manuscript highlighted in gray.

In this review, the authors reviewed studies on the rearrangement of centromere and telomere sequences, focusing specifically on the frequency and methodology used to detect such rare events in wild-type cells. The article indeed provides a review of the information for the latter study. However, the authors tend to mix pathological/experimental findings with physiological findings. Furthermore, some wording and statements are vague and require further polishing to make the article clearer.

Here are the general comments:

1.          Some similar terms are provided without an explanation of the difference. For example, gross chromosomal rearrangements (GCRs)/chromosomal aberrations (CAs) in line 58 (especially only GCRs are mentioned in the later section), and reproductive isolation/speciation in line 72. It needs to be addressed whether these terms have the same meaning or if there are differences between them.

We thank the reviewer for his/her close reading and insightful comments on our manuscript. Our usage of the terms “gross chromosomal rearrangements (GCRs)” and “chromosomal aberrations (CAs)” or “reproductive isolation’ and “speciation” in the introduction section of the previous manuscript was misleading without clear discrimination between them. To avoid confusion, we have omitted “chromosomal aberrations (CAs)” and “speciation” from the revised manuscript and corrected the sentences and follows:

Lines 56-59

These alterations are often catastrophic and are referred to collectively as gross chromosomal rearrangements (GCRs) [18,19]. Detachment and attachment of centromeres and telomeres occurs multiple times during GCRs [10,19,20] (Figure 1).

Lines 68-70

GCRs have been frequently observed in the malignant tumor cells [10,13,19,20]. Apart from catastrophes, GCR-like chromosomal alterations have also been implicated in reproductive isolation [22-25].

2.          In line 91, the authors state that "The special organization of telomeres is ensured by telomerase..." Telomerase is an enzyme that elongates telomere length but does not directly affect the structure of telomeres. Therefore, the relationship between the special organization of telomeres and telomerase needs to be described more clearly.

We thank the reviewer for this sharp comment. We agree with the reviewer that our statement on the contribution of telomerase to telomere structure was misleading in the previous manuscript. According to the reviewer’s suggestion, we have corrected the sentences in the section 2 of the revised manuscript as follows:

Lines 95-101

The capping function of telomeres is fulfilled through the action of specialized proteins recruited to the repeat sequences as well as through the formation of a higher order structure [5,14]. Telomere repeats are synthesized by telomerase, a specialized reverse transcriptase that elongates the 3’ end of telomeric DNA using a short region of its RNA subunit as a template. Telomerase action buffers against progressive telomere erosion caused by the incomplete replication of terminal DNA in each cell cycle and sustains the foundation for telomere structure [4,5].

3.          In lines 94-95, the authors state that "telomeres are subject to uncapping and telomere fusions occur readily when the telomere capping protein or the telomerase enzyme is functionally compromised, as reported in mammals and various yeast species." This statement may be imprecise and need to modified since not all cells in mammals express telomerase. Does a cell with no telomerase expression have a higher frequency of telomere fusion?

We agree with the reviewer that our description on the mutant condition in which telomere fusion can be detected was imprecise in the previous manuscript. We have corrected the sentences in the section 2 of the revised manuscript as follows:

Lines 101-104

Hence, telomeres are subject to uncapping and telomere fusions occur readily when the telomere capping protein or the DSB checkpoint signaling along with the telomerase enzyme is functionally compromised [36-44]. However, our concern is the occurrence of telomere fusions in an unchallenged, natural context.

4.          In this review, the authors emphasize the rare abnormal events that occur in wild-type cells. Yet, they focus on the frequency and methodology to detect them. The importance of these events must be highlighted in the article.

We believe that the flexible nature of chromosomes, especially the flexible attachment of centromeres and telomeres, is important. This was our motivation to prepare this article, the details of which are already described in the manuscript (e.g., lines 47-60 of the revised manuscript). The importance is independent of its frequency. We do not say it is important because of its high frequency nor its low frequency. In fact, how frequent the events representing chromosome flexibility are in wild-type context is a main topic of this article. In this respect, we have added a statement to the introduction of the revised manuscript as follows:

Lines 85-89

Instead, this review sheds light on the natural frequencies of events in the wild type. This allows a comparison of the efficiencies of different events in native context and permits combining these efficiencies in the future, which may be useful in obtaining an integrated perspective on chromosome flexibility (Figure 1).

5.          In Figure 1, the authors depict different chromosomal events with arrowheads. Is there a specific order in which these events occur? Moreover, the definition of the different events is not provided in the figure or text, which needs to be described. It is recommended to describe different events in different figure panels.

There is no specific order of occurrence in the events shown in Figure 1. The same outcome may be reached from the events in different orders. We have added this statement to the Figure 1 legend in the revised manuscript (lines 63-64). In response to the reviewer’s suggestion, we have also prepared separate figures indicating telomere fusion (Figure 2), centromere inactivation (Figure 4), telomere healing (Figure 5), and neocentromere formation (Figure 6) in relation to Figure 1 in the revised manuscript, as follows:

Lines 105-107

Figure 2. Telomere fusion event. It is also referred to as chromosome end-to-end fusion or telomere-to-telomere fusion. It occurs essentially when chromosome ends are recognized by cellular DSB repair pathways as DNA damages.

Lines 266-267

Figure 4. Centromere inactivation event. It results in the formation of dysfunctional acentric chromosome when it occurs to monocentric chromosome.

Lines 377-379

Figure 5. Telomere healing event. It is also referred to as chromosome healing or de novo telomere addition. It essentially stabilizes the broken DNA ends and protects them from DSB repair mechanisms.

Lines 509-510

Figure 6. Neocentromere formation event. It occurs to non-centromeric DNA regions epigenetically by stable incorporation and propagation of centromere-specific histone H3 variant CENP-A.

6.          In Table 1, the authors list the frequency of different chromosome events in wild-type cells. However, the table is confusing. First, the unit or definition of frequency is not mentioned (e.g., per cell, per chromosome, or cells per day). Second, it seems like the table includes both the frequency of natural conditions and the frequency after artificial treatment (e.g., neocentromere formation in C. albicans). The separation of these two different conditions is required since they are not compatible. 

We apologize for any confusion. Regarding the first point, the units of frequency in Table 1 are essentially per experimental setup designed or installed at a given locus on a single chromosome. We have added the unit statement to the Table 1 caption in the revised manuscript as shown below. Regarding the second point, the frequencies of the events shown in Table 1, not only for neocentromere formation in C. albicans, are all after artificial treatment if the experimental setup is considered artificial. The natural conditions we sticked to in this article mean that the treatment is performed on a wild-type background and not on a mutant. In this regard, there is no different condition in Table 1, as the reviewer suggested in his/her comment, and we have responded just the first point.

Lines 131-132

Table 1. Experimentally estimated frequencies of chromosomal events per given chromosome per cell in the wild-type context.

7.          In Figure 2, the authors represent different experimental setups. However, a detailed description of the design is lacking and needs to be added to the text. It is recommended to describe different experiments in different panels to make the figure clearer.

In response to the reviewer’s suggestion, we have added the description of the experimental design to the legend of Figure 3 (formerly Figure 2) in the revised manuscript as shown below. However, it is difficult to prepare appropriate individual panels showing each of the different experimental settings from Figure 3 (formerly Figure 2). This is because many of the pictures in Figure 3 (formerly Figure 2) symbolically represent individual settings used in different organisms. Once separated, more panels would have to be used to depict details representing the organism-specific differences, which would be space-consuming and also prevent readers from comparing experimental setups for different chromosomal events in a unified manner. Therefore, we have just added the description of the experimental design to the legend of Figure 2 (formerly Figure 2).

Lines 157-167

Figure 3. Representative experimental setups to capture spontaneous chromosomal alteration events. In every setup, coding regions of selectable, counter-selectable, or fluorescent marker genes (AG) are used appropriately in combination with appropriate promoters (P) in an appropriate arrangement to monitor the desired rearrangement by changes in gene expression. Essentially, minichromosome circularization assay in S. pombe utilizes RNA splicing, centromere deletion assay in S. pombe, S. cerevisiae or chicken and dicentric chromosome induction in S. pombe utilizes site-specific recombination, centromere inactivation assay in S. cerevisiae utilizes forced transcription, and dicentric chromosome induction assay in C. deuterogattii or S cerevisiae and centromere deletion assay in C. deuterogattii, S. cerevisiae or human utilize CRISPR-Cas9-mediated genome editing for selection. Byproducts are sometimes generated and are indicated by light colored arrows. The same assays performed in different organisms are shown together. See text for details.

8.          Some methodologies are vague, such as quantitative PCR-based experiments to detect telomere fusions (Line 217) and non-biased GCR assay for telomere healing (Line 363). Since the article aims to review the methodology, the experimental design needs to be described clearly.

We agree with the reviewer that the description of the PCR-based assay to detect telomere fusion was insufficient and the description of the GCR assay for telomere healing was not well structured in the previous manuscript. We have corrected these descriptions in the revised manuscript as follows:

Lines 109-112

A PCR-based single molecule assay using the primers designed for telomere-adjacent regions encompassing single-nucleotide polymorphisms and directed toward the chromosome terminus captured telomere fusions at a frequency of ~4 × 10−6 in telomerase-positive cultured human cells [45]

Lines 406-410

This was independently confirmed in S. cerevisiae using the non-biased GCR assay [108], which selects for the spontaneous loss of two counter-selectable marker genes located adjacent to each other on the terminal non-essential region of a chromosome and allows the isolation of various chromosomal rearrangements including telomere healing [65,109].

Round 2

Reviewer 1 Report

The authors have adequately addressed the reviewers concerns.

Author Response

Thanks for your kind review.